# LEARNING-AUGMENTED ALGORITHM FOR $k$-MEDIAN OF LINES CLUSTERING

## ABSTRACT

The line-based clustering is a natural extension of classical $k$-clustering problem with considerable applications in computer vision, missing data analysis, and related areas. Despite its practical importance, the unbounded nature of lines and the failure of the triangle inequality for point-to-line distances undermine the structural properties required for theoretical analysis, thereby causing that the theoretical foundation of line-based clustering remains far less developed than that of point-based clustering. In this paper, we study the $k$-median of lines problem, and address these challenges within the learning-augmented paradigm that leverages given auxiliary information in form of predicted labels to guide clustering. Specifically, we propose a new learning-augmented algorithm for the $k$-median of lines problem, in which auxiliary label is exploit to guide the sampling process, and the anchor set induced by line pairs is proposed to guarantees the inclusion of high-quality representative centers. Moreover, we theoretically prove that our proposed algorithm achieves a $(1 + O(\alpha))$-approximation in time $O\left(\frac{9^d}{(1-2\alpha)^2} n \log n \ln \frac{k}{\theta}\right)$ for the $k$-median of lines problem. In particular, in the low-dimensional Euclidean space, our algorithm can obtain a $(1 + O(\alpha))$-approximation with near-linear time complexity in the input size. Experimental results demonstrate that our algorithm consistently outperforms existing approaches in both solution quality and computational efficiency.

## 1 INTRODUCTION

Clustering is a fundamental research topic in computer science that has received significant attention over the past few decades. The central goal of clustering is to partition a given set of points into several disjoint clusters such that similar points end up in the same cluster, and dissimilar points are separated into different clusters (Pan et al. (2013)). Among various formulations, classical $k$-clustering problems such as $k$-means, $k$-median, and $k$-center have received significant attention due to their strong theoretical foundations and broad practical relevance.

A natural extension of the classical $k$-clustering problem is the $k$-clustering of lines problem, where the input consists of lines rather than points. This formulation arises naturally in domains such as computer vision (e.g., estimating object positions from multiple directional camera observations), robotics (e.g., clustering sensor beams or trajectories), and traffic analysis (e.g., grouping directional flows based on road orientation or motion vectors). Despite its practical significance, the study of line-based clustering remains comparatively underdeveloped, both in terms of theoretical guarantees and algorithmic efficiency. The main obstacles arise from the unbounded nature of lines and the fact that point-to-line distance violates the triangle inequality, which together undermine the structural properties essential for theoretical analysis.

One of the most popular recent approaches for overcoming these computational barriers is to exploit the auxiliary structure often present in real-world inputs, such as similarity to previously seen instances, which can be leveraged to guide and improve clustering outcomes. More generally, recent algorithms often create a (possibly erroneous) predictor by leveraging auxiliary information or clusterings of related datasets, which helps to compensate for the structural difficulties inherent to the problem Aamand et al. (2025); Grigorescu et al. (2024); Gamlath et al. (2022); Ashtiani et al. (2016); Kraska et al. (2018); Mitzenmacher (2018); Huang et al. (2025).

Table 1: Approximation Results for the $k$-median of lines problem with (without) learning-augmented setting.

| Euclidean space | Approximation | Time | Reference |
|:---:|:---:|:---:|:---:|
| $\mathbb{R}^2$ | $1 + \epsilon$ | $O(n^{2k} \log n))$ | Perets (2011) |
| Any dimension | $1 + \epsilon$ | $O((\frac{1}{\epsilon})^{\text{poly}(k)} n (\log n)^{\text{poly}(k)})$ | Perets (2011) |
| Low dimension | $1 + O(\alpha)$ | $O(\frac{1}{(1-2\alpha)^2} n \log n \ln \frac{k}{\theta})$ | Theorem 1 |
| Any dimension | $1 + O(\alpha)$ | $O(\frac{9^d}{(1-2\alpha)^2} n \log n \ln \frac{k}{\theta})$ | Theorem 2 |

In this paper, we study the $k$-median of lines problem, which extends the well-studied $k$-median formulation from discrete point sets to continuous line inputs in $\mathbb{R}^d$. More formally, given a set $L$ of lines in $\mathbb{R}^d$ and an integer $k$, the goal is to find $k$ centers $C$ in $\mathbb{R}^d$ that minimize $\Delta(L, C) = \sum_{\ell \in L} \min_{c \in C} d(\ell, c)$, where the distance between a line $\ell$ and a point $c$ is defined as $d(\ell, c) = \min_{x \in \ell} \|x - c\|_2$. We further study this problem in the learning-augmented setting, where each line is equipped with a (possibly erroneous) predicted cluster label that can be exploited to guide the clustering process.

**The learning-augmented setting.** In the learning-augmented setting, an auxiliary clustering $\{L_1, \ldots, L_k\}$ of $L$ is provided with a label error rate $\alpha$, where all points in $L_i$ share label $i$ for any $i \in [k]$. Let $\{L_1^*, \ldots, L_k^*\}$ denote an optimal clustering of $L$. In this setting, for each $i \in [k]$, we have that $|L_i \cap L_i^*| \geq (1 - \alpha) \max(|L_i|, |L_i^*|)$. Following Nguyen et al. (2023), the error rate $\alpha$ is assumed to be less than $1/2$. For notational simplicity, we define $m_i = |L_i|$ and $m_i^* = |L_i^*|$ for every $i \in [k]$.

## 1.1 MOTIVATION FOR OUR WORK

The $k$-clustering of lines can be regarded as a natural generalization of the classical $k$-clustering of points, which has found extensive applications across numerous domains. In the following, we highlight representative examples from different research areas.

**Computer vision.** A fundamental application of line-based clustering arises in computer vision (e.g., Marom & Feldman (2019); Lotan et al. (2022)), where directional observations naturally induce sets of lines that can be clustered to recover underlying structures. For example, when multiple cameras observe fixed objects, each camera yields a directional observation toward the object, which can be represented as a line. Clustering such lines allows us to infer the geometric intersection structure, thereby estimating the positions of the underlying targets.

**Handling missing data.** Missing data analysis constitutes a fundamental application of line-based clustering (e.g., Braverman et al. (2021); Lotan et al. (2022)). In this application, each record in the input data is represented by a point in $d$-dimensional space. When an entry is missing, it can be replaced by all possible values of that attribute, thereby transforming the point into a line in $\mathbb{R}^d$ that is parallel to one of the coordinate axes. Clustering these lines allows the algorithm to effectively handle datasets with missing attributes, while systematically inferring principled estimates for the absent values. For example, in a medical dataset, it can still uncover potential patient groups despite missing blood pressure records and provide representative estimates of the missing values.

## 1.2 OUR CONTRIBUTIONS

In this paper, we propose a sampling method to overcome the aforementioned obstacles. Our main contribution provides a positive answer to the stated question, summarized as follows.

- We propose a new technique, called Anchor Set, to construct a bounded region defined by a pair of sampled lines, enabling the selection of representative sampling points or candidate centers.
- Building on this technique, we develop a sampling algorithm that avoids reliance on the triangle inequality and efficiently constructs a candidate center set with provable approximation guarantees.

- We obtain $(1 + O(\alpha))$-approximation result for the $k$-median of lines problem in low-dimensional Euclidean space, running with near-linear time in the data size.

We summarize the results in the literature and ours in Table 1. Formally, we have following result for the $k$-median of lines problem problem.

**Theorem 1.** *There is a randomized learning-augmented algorithm that achieves a $(1 + O(\alpha))$-approximation in time $O(\frac{1}{(1-2\alpha)^2} n \log n \ln \frac{k}{\theta})$ for the $k$-median of lines problem in low-dimensional Euclidean space.*

**Theorem 2.** *There is a randomized learning-augmented algorithm that achieves a $(1 + O(\alpha))$-approximation in time $O(\frac{9^d}{(1-2\alpha)^2} n \log n \ln \frac{k}{\theta})$ for the $k$-median of lines problem in high-dimensional Euclidean space.*

### 1.3 RELATE WORK

**Line-based clustering.** Gao et al. (2008) studied the 1-line center problem of lines in $\mathbb{R}^d$, where the goal is to identify a single center minimizing the maximum distance to a set of input lines. They presented the first coreset construction for this setting, computing an $\varepsilon$-coreset of size $O(1/\varepsilon)$ in time $O(nd\text{poly}(1/\varepsilon))$. Extending this line of research, Gao et al. (2010) proposed a $(2 + \varepsilon)$-approximation algorithm for the $k$-center of lines problem in $\mathbb{R}^2$ when $k = 2$ or $k = 3$. Their algorithm runs in quasi-linear time with respect to both the number of lines and the dimensionality of the input space, demonstrating that strong approximation guarantees can be achieved efficiently for small values of $k$. In parallel, Ommer & Malik (2009) initiated the study of the $k$-median of lines problem. They introduced a heuristic algorithm that aims to minimize the sum of distances from lines to their nearest centers. However, their method lacks formal approximation guarantees and does not provide a bounded runtime complexity, leaving open the theoretical understanding of the problem. Perets (2011) developed an approximation algorithm for the $k$-median of lines in $\mathbb{R}^2$ by first computing a bi-criteria approximate solution, and then applying coreset-based refinement. The resulting algorithm achieves a $(1+\varepsilon)$-approximation with a running time of $O(n(\log n/\varepsilon))^{O(k)}$, for any $\varepsilon > 0$, thus providing the first provable guarantees for $k$-median clustering of lines in low dimensions. More recently, Marom & Feldman (2019) addressed the more challenging $k$-means of lines problem and proposed a coreset construction that yields an $\varepsilon$-coreset of size $O(dk^{O(k)} \log n/\varepsilon^2)$. Their algorithm runs in time $O(d^3 n \log(n)k \log k + (d/\varepsilon)^2 + ndk^{O(k)})$, marking a significant advancement in scalable approximation schemes for clustering line-based data. Lotan et al. (2022) studied the $k$-median problem on sets of lines and proposed an $\varepsilon$-coreset of size $O(\log^2 n)$. They achieves a construction time of $O(n \log n)$, providing an efficient data reduction technique with theoretical guarantees.

**Learning-augmented algorithms for clustering.** Recently, to develop more practical approximation algorithms for classical $k$-clustering problem, a series of studies has focused on algorithms augmented with predictions Gamlath et al. (2022); Ashtiani et al. (2016); Kraska et al. (2018); Mitzenmacher (2018); Huang et al. (2025). Specifically, Gamlath et al. (2022) studied the $k$-means problem augmented with a predictor that provides additional noisy labels, where the labels are generated through random perturbations of an optimal clustering solution. The predictor is parameterized by an error rate $\alpha \in [0, 1)$, such that the symmetric difference between each predicted cluster and its corresponding optimal cluster is at most an $\alpha$-fraction of the optimal cluster size. Based on this model, Gamlath et al. (2022) developed a $(1 + O(\alpha))$-approximation algorithm with polynomial running time for fixed $k$ and $d$, under the condition that the error rate $\alpha < 1/4$. Ergun et al. (2022) proposed an alternative learning-augmented clustering model, where the predictor is characterized by an error rate $\alpha \in [0, 1)$, ensuring that each predicted cluster contains at most an $\alpha$-fraction of false positives and false negatives. Based on this model, they developed a $(1+O(\alpha))$-approximation algorithm with near-linear runtime in the dataset size. Nguyen et al. (2023) proposed a deterministic algorithm for the $k$-means problem in the learning-augmented setting, which achieves improved clustering cost bounds compared to prior randomized approaches, while maintaining a near-optimal runtime of $O(dn \log n)$. For the $k$-median problem, they further enhanced existing results by achieving a biquadratic improvement in the dependence of the approximation factor on the prediction error rate $\alpha$, obtaining a $(1 + O(\alpha))$-approximation with an efficient runtime of $O(nd \log^3 n/\alpha)$. Huang et al. (2025) built on the work of Ergun et al. by proposing sampling-based algorithms guided by predicted labels, achieving a $(1 + O(\alpha))$-approximation in linear time for large-scale clustering.

## 2 PRELIMINARIES

For any positive integer $m \in \mathbb{N}^{\geq 1}$, let $[m] = \{1, \dots, m\}$. In this paper, we consider the $k$-median of line clustering over a set $L$ of $n$ lines. Notably, we assume that all lines in $L$ are pairwise non-parallel (possibly intersecting or skew), ensuring distinct directional relationships. Indeed, our algorithm remains robust to parallel lines with minimal impact on solution quality, as similarly addressed in Perets (2011).

Given any two points $p, q \in \mathbb{R}^d$, let $\|p - q\|$ denote the Euclidean distance between $p$ and $q$. Given a set $L$ of lines in $\mathbb{R}^d$ and a set $C \subseteq \mathbb{R}^d$ of points, for any $\ell \in L$ and $c \in C$, let $\delta(\ell, c) = \min_{p \in \ell} \|p - c\|$ denote the distance between $\ell$ and $c$, which is the shortest distance from any point on the line $\ell$ to $c$, and let $\pi_\ell(c)$ denote the projection of $c$ on $\ell$. Let $\delta(\ell, C) = \min_{c \in C} \delta(\ell, c)$. Further, we define the clustering cost of $L$ with respect to $C$ as $\Delta(L, C) = \sum_{\ell \in L} \delta(\ell, C)$. Formally, the $k$-median of lines (denoted as $k$-ML) problem considered in this paper can be defined as follows.

**Definition 1** (the $k$-median of lines problem). *Given a set $L \subseteq \mathbb{R}^d$ of lines in a $d$-dimensional Euclidean space and a positive integer $k$, the goal is to find a set $C \subset \mathbb{R}^d$ of $k$ centers such that the objective $\Delta(L, C)$ is minimized.*

Given an instance $(L, d, k)$ of the $k$-median of lines problem, $C$ is called a feasible solution of this instance if $C \subseteq \mathbb{R}^d$ is a set of $k$ points. Throughout this paper, let $C^* = \{c_1^*, \dots, c_k^*\}$ denote the optimal solution, and let $\{L_1^*, \dots, L_k^*\}$ denote the corresponding optimal clusters by assigning each line in $L$ to its closest centers in $C^*$. Let $\tau = \Delta(L, C^*)$ be the cost of optimal solution. For any $i \in [k]$, let $\tau_i = \sum_{\ell \in L_i^*} \delta(\ell, c_i^*)$. Thus, we have $\sum_{i=1}^k \tau_i = \tau$.

To leverage the prediction more tightly, we introduce a mild stability condition reminiscent of perturbation resilience, called $\alpha$-stability.

**Definition 2** ($\alpha$-stability). *Given a partition $\{L_1, \dots, L_k\}$ of $L$ with a label error rate $\alpha$, the partition is called $\alpha$-stability if for any $i \in [k]$ and any subset $L_i^\alpha \subseteq L_i$ with $|L_i^\alpha| \geq \lceil (1 - \alpha) m_i \rceil$, the following holds. Let $c^\alpha$ be an optimal 1-median of $L_i^\alpha$, and let $c_i^*$ be the optimal center for $L_i^*$. Then, for any line $\ell \in L_i^\alpha \cap L_i^*$, $\|c^\alpha - c_i^*\| \leq \delta(c^\alpha, \ell) + \delta(c_i^*, \ell)$.*

Intuitively, the $\alpha$-stability assumption require that each auxiliary cluster has at least the $\lceil 1 - \alpha \rceil$ proportion of lines closed to the optimal center. Some similar assumptions have been proposed in literatures Ding (2021); Awasthi et al. (2012); Balcan et al. (2020); Balcan & Liang (2016).

## 3 THE LEARNING-AUGMENTED ALGORITHM FOR THE $k$-MEDIAN OF PROBLEM

### 3.1 THE OVERVIEW OF OUR ALGORITHM

The learning-augmented algorithm for the $k$-median problem, as developed by Nguyen et al. (2023), begins by selecting an anchor point from each predicted cluster to serve as a proxy for the unknown optimal center. Unlike the classical $k$-median clustering problem, which takes as input a finite set of points, the $k$-median of lines problem considers a set of lines in $\mathbb{R}^d$ as input, making the clustering task significantly more challenging due to the continuous and unbounded nature of lines. Therefore, it is necessary to explicitly construct a candidate point set that captures the underlying geometry of each predicted cluster, even under noisy or partially incorrect labels. Moreover, the distance between a line and a point does not satisfy the triangle inequality, which breaks many structural properties crucial for theoretical analyses.

To overcome these challenges, we develop a new sampling algorithm for the $k$-median of lines problem based on the learning-augmented setting, referred to LA-$k$ML, which is presented in Algorithm 1. We now provide an intuitive overview of our algorithm for the $k$-median of lines problem. Given an instance $(L, d, k)$ of the $k$-median of lines problem, an auxiliary clustering $\{L_1, \dots, L_k\}$ of $L$ with a label error rate $\alpha < 1/2$, our algorithm works with each auxiliary cluster $L_i$ $(i \in [k])$, independently. Specifically, consider an auxiliary cluster $L_i$. Instead of sampling a point from $L_i$ directly, we first construct an anchor set $\mathcal{M}$ based on two lines sampled from $L_i$ randomly, and then pick a point $x \in \mathcal{M}$ randomly. We next remove the farthest $\lceil \alpha n_i \rceil$ lines in $L_i$ from $x$. For the remaining lines in $L_i$, we compute the geometric median. To obtain a better center, for each auxiliary

---

**Algorithm 1** LA-$k$ML

---

**Input**: An instance $(L, d, k)$ of the $k$-ML problem, an auxiliary clustering $\{L_1, \ldots, L_k\}$ of $L$ with a label error rate $\alpha < 1/2$, and a parameter $\theta$

**Output**: A set $\hat{C} = \{\hat{c}_1, \ldots, \hat{c}_k\} \subset \mathbb{R}^d$ of at most $k$ centers

1: **for** $i = 1$ **to** $k$ **do**
2:     $R \leftarrow O(\frac{1}{(1-2\alpha)^2} \ln \frac{k}{\theta})$;
3:     **for** $j = 1$ **to** $R$ **do**
4:         Sample two lines $\ell_1, \ell_2$ from $L_i$ randomly and independently;
5:         $r \leftarrow$ compute an upper bound based on $\ell_1, \ell_2$;
6:         Construct the anchor set $\mathcal{M}$ based on $\ell_1, \ell_2, r$;
7:         $x \leftarrow$ sample a point from $\mathcal{M}$ randomly;
8:         $L_i' \leftarrow$ the set of $\lceil \alpha n_i \rceil$ lines in $L_i$ farthest from $x$;
9:         $\hat{c}_i^j \leftarrow$ the median of $L_i \setminus L_i'$;
10:     **end for**
11:     Let $\hat{c}_i$ be the $\hat{c}_i^j$ with minimum cost;
12: **end for**
13: **return** $\hat{C} = \{\hat{c}_1, \ldots, \hat{c}_k\}$.

---

cluster $L_i$, we repeat the above process $R$ times, and return the center with minimum cost. Finally, by collecting the center for each $i \in [k]$, we obtain the resulting solution $\{\hat{c}_1, \ldots, \hat{c}_k\}$.

### 3.2 LOW-DIMENSIONAL EUCLIDEAN SPACE

To facilitate understanding of our learning-augmented setting, this section focuses on the sampling process under the guidance of predicted cluster labels. Specifically, for each predicted cluster, we generate a candidate point set using the *Anchor Set* structure, which is constructed by sampling two lines independently from the same predicted cluster with a certain probability and computing their geometric intersection region. A representative point is then sampled from this anchor set to serve as a proxy for the optimal cluster center. If the predicted clustering has a low error rate (i.e., $\alpha < 1/2$), we show that the anchor set is likely to contain a point close to the optimal center with high probability. In the section, we formally prove that this construction leads to a $(1 + O(\alpha))$-approximate solution with near-linear time complexity in the input size.

**Lemma 3.** *Consider an instance $(L, d, k)$ of the $k$-ML problem and an auxiliary clustering $\{L_1, \ldots, L_k\}$ of $L$ with a label error rate $\alpha$. For any $i \in [k]$, if a line $\ell \in L_i$ is sampled uniformly at random, then with probability at least $\frac{1-2\alpha}{2}$, it holds that $\delta(\ell, c_i^*) \leq \frac{2\tau_i}{n_i}$*

*Proof.* For any $i \in [k]$, let $Q_i = L_i \cap L_i^*$. By the definition of learning-augmented setting, we have $|Q_i| \geq (1 - \alpha) \max\{|L_i|, |L_i^*|\} \geq (1 - \alpha) n_i$. Since $|Q_i| \leq |L_i^*|$, it directly follows that $\Delta(Q_i, c_i^*) \leq \Delta(L_i^*, c_i^*) = \tau_i$. Thus, if we sample a line $\ell$ from $L_i$ randomly, the expectation of the distance between $\ell$ and $c_i^*$ satisfies $\mathbb{E}_{\ell \sim Q_i}[\delta(\ell, c_i^*)] = \frac{1}{|Q_i|} \cdot \Delta(Q_i, c_i^*) \leq \frac{\tau_i}{|Q_i|} \leq \frac{\tau_i}{(1-\alpha)n_i}$. Applying Markovs inequality, we obtain $\Pr\left(\delta(\ell, c_i^*) > \frac{(2-2\alpha)\tau_i}{(1-\alpha)n_i} \mid \ell \in Q_i\right) \leq \frac{\frac{\tau_i}{(1-\alpha)n_i}}{\frac{(2-2\alpha)\tau_i}{(1-\alpha)n_i}} = \frac{1}{2-2\alpha}$. Thus, the complementary event has probability at least $\Pr\left(\delta(\ell, c_i^*) \leq \frac{(2-2\alpha)\tau_i}{(1-\alpha)n_i} \mid \ell \in Q_i\right) \geq 1 - \frac{1}{2-2\alpha} = \frac{1-2\alpha}{2-2\alpha}$. Considering the conditional probability and noting $\Pr(\ell \in Q_i) = \frac{|Q_i|}{|L_i|} \geq 1 - \alpha$, we have $\Pr\left(\delta(\ell, c_i^*) \leq \frac{(2-2\alpha)\tau_i}{(1-\alpha)n_i}\right) \geq \frac{1-2\alpha}{2-2\alpha} \cdot \Pr(\ell \in Q_i) \geq \frac{1-2\alpha}{2}$. Hence, with probability at least $\frac{1-2\alpha}{2}$, we have $\delta(\ell, c_i^*) \leq \frac{2\tau_i}{n_i}$. $\square$

Lemma 3 implies that if we sample two line $\ell_2, \ell_2 \in L_i$ independently and randomly, then by the union bound, both two lines are within distance $\frac{2\tau_i}{n_i}$ from $c_i^*$ with probability at least $\frac{(1-2\alpha)^2}{4}$.

The previous method in Nguyen et al. (2023) samples directly a point from the auxiliary cluster to approximate the optimal center, which is hard to extended to the $k$-median of lines problem due

to its input as lines. To address this issue, our improved strategy is to sample a point from an anchor set that can be constructed based on two lines sampled from auxiliary cluster randomly and independently. To proceed, we introduce the notion of an anchor set as follows.

**Definition 3** (Anchor Set). *Let $\ell_1$ and $\ell_2$ be two intersecting lines in $\mathbb{R}^2$, intersecting at a point $p = \ell_1 \cap \ell_2$. For each line $\ell_i$ ($i = 1, 2$), we construct two lines parallel to $\ell_i$ by shifting it along its normal direction by $\pm r$. Let $\mathcal{L}_1 = \{\ell_1^-, \ell_1, \ell_1^+\}$ and $\mathcal{L}_2 = \{\ell_2^-, \ell_2, \ell_2^+\}$ denote the sets of three parallel lines derived from $\ell_1$ and $\ell_2$, respectively. The Anchor Set is defined as the set of all pairwise intersections between lines in $\mathcal{L}_1$ and lines in $\mathcal{L}_2$: $\mathcal{A}_r(\ell_1, \ell_2) = \{\ell_i \cap \ell_j \mid \ell_i \in \mathcal{L}_1, \ell_j \in \mathcal{L}_2\}$. This construction yields a $3 \times 3$ grid of points forming a regular parallelogram pattern centered at the intersection point $p$.*

Let $\ell_1, \ell_2$ denote the two lines returned by step 4 of Algorithm 1. For any $i \in [k]$, we first compute the median $m_i$ of $L_i$ using the following result.

**Theorem 4** (Lemma 4.2.3 of Perets (2011)). *The 1-Median of lines problem can be solved in time $O(n \log n)$, where $n$ is the number of input lines.*

Let $r = 2\Delta(L_i, m_i)$. By above definition, we construct an anchor set $\mathcal{A}_r(\ell_1, \ell_2)$ based on $\ell_1, \ell_2, r$. Let $\mathcal{M} = \mathcal{A}_r(\ell_1, \ell_2)$. Let $x$ denote the point sampled from $\mathcal{M}$ randomly.

**Lemma 5.** *Consider an instance $(L, d, k)$ of the $k$-median of lines problem and an auxiliary clustering $\{L_1, \ldots, L_k\}$ of $L$ with a label error rate $\alpha$. For any $i \in [k]$, with probability $\frac{t(1-2\alpha)^2}{4}$, we have $\|x - c_i^*\| \le \frac{2\tau_i}{n_i}$, where $t$ is a constant.*

*Proof.* We first show that there must exist a point in $\mathcal{M}$ within distance $\frac{2\tau_i}{n_i}$ from $c_i^*$. It is easy to follow that the number of points in the anchor set $\mathcal{M}$ is $9^d$, assuming that all such pairs intersect. By the above discussion, with probability at least $\frac{(1-2\alpha)^2}{4}$, both two lines $\ell_2, \ell_2$ are within distance $\frac{2\tau_i}{n_i}$ from $c_i^*$. By the process of constructing the anchor set $\mathcal{M}$, we only to show that the value of $\frac{2\tau_i}{n_i}$ is upper bounded by $r$, i.e., $\frac{2\tau_i}{n_i} \le r$. Recall that in Lemma 3, we have $Q_i > \frac{n_i}{2}$, i.e., $L_i$ contains at least half of points in $L_i^*$. Therefore, we have $2\Delta(L_i, m_i) \ge \tau_i$. Further, $r \ge \frac{2\tau_i}{n_i}$. If we pick a point $x$ from $\mathcal{M}$ randomly, the with probability at least $\frac{1}{9^d} \cdot \frac{(1-2\alpha)^2}{4} = \frac{t(1-2\alpha)^2}{4}$, $\|x - c_i^*\| \le \frac{2\tau_i}{n_i}$, where $t$ is a constant. $\square$

For brevity, we define the event $\mathcal{E}_i$ as follows: for any $i \in [k]$, we have $\|x - c_i\| \le \frac{2\tau_i}{n_i}$. Additionally, for any $i \in [k]$, we define the sets of false negatives and false positives as $A_i = L_i' \cap L_i^*$ and $B_i = L_i \setminus (L_i' \cup L_i^*)$, respectively. The following lemma relates these two sets.

**Lemma 6.** *For any $i \in [k]$, we have $L_i \setminus L_i' = ((L_i \cap L_i^*) \setminus A_i) \cup B_i$ (see Appendix A.1 for the proof).*

For any $i \in [k]$ and $j \in [R]$, let $\hat{c}_i^j$ denote the median of the clipped set $L_i \setminus L_i'$ computed by Theorem 4. We now show that the clustering cost of the clipped set $L_i \setminus L_i'$ is close to that of the optimal cluster $L_i^*$.

**Lemma 7.** *Conditioned on $\mathcal{E}_i$ ($i \in [k]$), for any $j \in [R]$, we have $\Delta(L_i \setminus L_i', \hat{c}_i^j) \le (1 + 8\alpha)\tau_i$.*

*Proof.* By Lemma 6, we have $\Delta(L_i \setminus L_i', c_i^*) = \Delta(L_i \cap L_i^*, c_i^*) - \Delta(A_i, c_i^*) + \Delta(B_i, c_i^*)$. Again, from Lemma 6, it follows that $L_i \setminus L_i' = ((L_i \cap L_i^*) \setminus A_i) \cup B_i$, where clearly $A_i \subset L_i \cap L_i^*$ and $B_i \cap (L_i \cap L_i^*) = \emptyset$. Thus, we obtain the following cardinality relation $|L_i \setminus L_i'| = |L_i \cap L_i^*| - |A_i| + |B_i|$. Since $|L_i \setminus L_i'| \le (1 - \alpha)|L_i|$ and $|L_i \cap L_i^*| \ge (1 - \alpha)|L_i|$, we get that $|B_i| \le |A_i| \le \alpha n_i$. By definition, each line in $A_i$ is farther from point $x$ than each line in $B_i$. Since $|B_i| \le |A_i|$, we associate each line $\ell \in B_i$ uniquely with a corresponding line $u_\ell \in A_i$, ensuring $\delta(x, \ell) \le \delta(x, u_\ell)$. Using the triangle inequality, for any line $\ell \in B_i$, we have

$$\delta(c_i^*, \ell) \le \|c_i^* - x\| + \delta(x, \ell) + \|\pi_\ell(c_i^*) - \pi_\ell(x)\| \le \|c_i^* - x\| + \delta(x, u_\ell) + \|\pi_\ell(c_i^*) - \pi_\ell(x)\|$$
$$\le 2\|c_i^* - x\| + \delta(x, u_\ell),$$
(1)

where the second inequality follows from the choice of $u_\ell$, and the third from the fact that the distance between the projections of two points onto a line is at most their original Euclidean distance. Similarly, again by the triangle inequality, we have

$$\delta(x, u_\ell) \leq \|x - c_i^*\| + \delta(c_i^*, u_\ell) + \|\pi_{u_\ell}(x) - \pi_{u_\ell}(c_i^*)\| \leq 2\|x - c_i^*\| + \delta(c_i^*, u_\ell). \quad (2)$$

Combining Equations (1) and (2), and invoking Lemma 5, we can obtain $\delta(c_i^*, \ell) \leq \delta(c_i^*, u_\ell) + 4\|c_i^* - x\| \leq \delta(c_i^*, u_\ell) + \frac{8\tau_i}{n_i}$. Summing over all lines in $B_i$, we get that $\Delta(B_i, c_i^*) = \sum_{\ell \in B_i} \delta(\ell, c_i^*) \leq \sum_{\ell \in B_i} (\delta(c_i^*, u_\ell) + \frac{8\tau_i}{n_i}) \leq \Delta(A_i, c_i^*) + 8\alpha\tau_i$ where the last inequality follows from $|B_i| \leq \alpha n_i$. Thus,

$$\Delta(L_i \setminus L_i', c_i^*) = \Delta((L_i \cap L_i^*) \setminus A_i, c_i^*) + \Delta(B_i, c_i^*) = \Delta(L_i \cap L_i^*, c_i^*) - \Delta(A_i, c_i^*) + \Delta(B_i, c_i^*)$$
$$\leq \Delta(L_i \cap L_i^*, c_i^*) + 8\alpha\tau_i \leq (1 + 8\alpha)\tau_i,$$

since clearly $\Delta(L_i \cap L_i^*, c_i^*) \leq \Delta(L_i^*, c_i^*) = \tau_i$. Finally, let $\hat{c}_i^j$ denote the median of the set $L_i \setminus L_i'$, and we immediately have $\Delta(L_i \setminus L_i', \hat{c}_i^j) \leq \Delta(L_i \setminus L_i', c_i^*) \leq (1 + 8\alpha)\tau_i$. $\qquad\square$

To boost the constant success probability of sampling a near-optimal point from the *Anchor Set* (event $\mathcal{E}_i$), the procedure is repeated $R$ times, and the estimate $\hat{c}_i$ yielding the lowest cost on the corresponding clipped subset is returned.

**Lemma 8.** *Let $R = \frac{4}{t(1-2\alpha)^2} \ln \frac{k}{\theta}$. Then, after $R$ repetitions, with probability at least $1 - \frac{\theta}{k}$, we have $\Delta(L_i \setminus L_i', \hat{c}_i) \leq (1 + 8\alpha)\tau_i$ (see Appendix A.2 for the proof).*

Lemma 8 establishes that, with a sufficient number of repetitions, the algorithm can identify an approximate center $\hat{c}_i$ whose cost is close to optimal with high probability, even in the presence of label noise with error rate $\alpha < 1/2$. This Lemma constitutes a key component in establishing the overall approximation guarantees of the algorithm.

**Lemma 9.** *If $\Delta(L_i \setminus L_i', \hat{c}_i) \leq (1 + 8\alpha)\tau_i$, then it holds that $\|\hat{c}_i - c_i^*\| \leq \frac{(2 + 8\alpha)\tau_i}{(1 - 2\alpha)n_i}$ (see Appendix A.3 for the proof).*

Lemma 9 shows that if the estimated center yields a near-optimal clustering cost on the clipped subset, then it is also guaranteed to be geometrically close to the true optimal center. This Lemma connects cost-based approximation with geometric accuracy under auxiliary labels.

**Lemma 10.** *With probability at least $1 - \frac{\theta}{k}$, the following inequality holds $\Delta(L_i \cap L_i^*, \hat{c}_i) \leq \Delta(L_i \cap L_i^*, c_i^*) + \frac{(8\alpha + 32\alpha^2)\tau_i}{(1 - 2\alpha)}$ (see Appendix A.4 for the proof).*

Lemma 10 shows that, with high probability, the estimated center achieves a clustering cost on the correctly labeled subset that is close to that of the optimal center, with only a small additive error due to label noise. Lemma 10 highlights the robustness of the algorithm in the presence of imperfect auxiliary labels.

**Lemma 11.** *With probability at least $1 - \frac{\theta}{k}$, $\Delta(L_i^*, \hat{c}_i) \leq (1 + \frac{12\alpha + 40\alpha^2 - 32\alpha^3}{(1-\alpha)(1-2\alpha)})\tau_i$ (see Appendix A.5 for the proof).*

We now consider our cost bound, success probability and run-time guarantees over all auxiliary clusters.

**Lemma 12.** *Given an instance $(L, d, k)$ of the $k$-ML problem, an auxiliary clustering $\{L_1, \ldots, L_k\}$ of $L$ with a label error rate $\alpha < 1/2$, and a parameter $\theta$, with probability at least $1 - \theta$, Algorithm 1 returns a set $\hat{C} \subset \mathbb{R}^d$ of centers such that $\Delta(L, \hat{C}) \leq (1 + O(\alpha))\tau$, where $O(\alpha) = \frac{12\alpha + 40\alpha^2 - 32\alpha^3}{(1-\alpha)(1-2\alpha)}$. Moreover, the running time of Algorithm 1 is $O(\frac{1}{(1-2\alpha)^2} n \log n \ln \frac{k}{\theta})$.*

*Proof.* Let $\hat{C} = \{\hat{c}_1, \ldots, \hat{c}_k\} \subset \mathbb{R}^d$ denote the set of $k$ centers returned by Algorithm 1. By Lemma 11, for any $i \in [k]$, $\Pr\left(\Delta(L_i^*, \hat{c}_i) \leq (1 + \frac{12\alpha + 40\alpha^2 - 32\alpha^3}{(1-\alpha)(1-2\alpha)})\tau_i\right) \geq 1 - \frac{\theta}{k}$. Let $O(\alpha) = \frac{12\alpha + 40\alpha^2 - 32\alpha^3}{(1-\alpha)(1-2\alpha)}$. Applying the union bound over the $k$ clusters gives

$\Pr\left(\sum_{i=i}^{k} \Delta(L_i^*, \hat{c}_i) \leq (1 + O(\alpha)) \sum_{i=i}^{k} \tau_i\right) \geq 1 - \theta$. Since $L = \cup_{i=1}^{k} L_i^*$ and $\tau = \sum_{i=1}^{k} \tau_i$, the desired cost bound follows.

We now bound the running time of Algorithm 1. For any $i \in [k]$, the 1-Median subroutine is executed for $R = O(\frac{1}{(1-2\alpha)^2} \ln \frac{k}{\theta})$ iterations. During one iteration on the auxiliary cluster $L_i$ of size $n_i$, we spend $O(n_i)$ time to compute distances, $O(n_i \log n_i)$ time to sort them, and another $O(n_i \log n_i)$ time to compute the clipped median (Theorem 4), i.e., $O(n_i \log n_i)$ in total. Thus, the running time on auxiliary cluster $L_i$ is $O(Rn_i \log n_i)$. Summing over all $k$ auxiliary clusters yields $\sum_{i=i}^{k} O(Rn_i \log n_i) = O(\frac{1}{(1-2\alpha)^2} n \log n \ln \frac{k}{\theta})$. $\qquad\square$

By Lemma 12, Theorem 1 can be proved.

### 3.3 EXTENDED TO HIGH-DIMENSIONAL

In this section, we describe how our algorithm can be extended to the general setting in high-dimensional Euclidean space. Unlike in $\mathbb{R}^2$, where any two non-parallel lines intersect, the presence of skew lines in higher dimensions renders the original Anchor Set insufficient, as lines may fail to intersect and can reside in different affine subspaces. To address this issue, we generalize the Anchor Set construction to better capture high-dimensional geometric relationships, while preserving the key coverage properties necessary for our approximation guarantees. We now formally redefine the Anchor Set for high-dimensional settings.

**Definition 4** (Anchor Set). *Consider two non-parallel lines $\ell_1, \ell_2 \subset \mathbb{R}^d$ and a parameter $r > 0$. For each $i \in \{1, 2\}$, we define a set of axis-aligned shifted lines as $\mathcal{L}_i = \{\ell_i^{\vec{s}} := \ell_i + \sum_{j=1}^{d} s_j \cdot r \cdot e_j \mid \vec{s} \in \{-1, 0, 1\}^d\}$, where $e_j$ is the unit vector along the $j$-th coordinate axis. We define the* anchor set $\mathcal{A}_r(\ell_1, \ell_2)$ *as the collection of all pairwise intersection points between lines in $\mathcal{L}_1$ and $\mathcal{L}_2$, i.e., $\mathcal{A}_r(\ell_1, \ell_2) = \{\ell \cap \ell' \mid \ell \in \mathcal{L}_1, \ell' \in \mathcal{L}_2, \ell \cap \ell' \neq \emptyset\}$.*

Building on generalized Anchor Set construction, we extend Algorithm 1 to handle lines in high-dimensional Euclidean space, and analyze its success probability and overall running time in this general setting. Specifically, we examine how the success probability and total runtime scale with the input dimension $d$. For each $i \in [k]$, we also define an event $\mathcal{E}_i$ corresponding to the successful selection of a representative center for cluster $i$. In the high-dimensional setting, the number of translated lines in the Anchor Set construction increases exponentially with $d$, resulting in a total of $3^d$ shifted versions of each sampled line. Consequently, the number of intersection points between two such families is bounded by $9^d$. Since each input line is correctly labeled with probability at least $1 - \alpha$, and the predictor makes independent errors, we show that the joint event $\mathcal{E}_i$ occurs with probability at least $\frac{(1-2\alpha)^2}{4 \cdot 9^d}$.

To ensure that all $k$ clusters are correctly represented with high probability, specifically with probability at least $1 - \frac{\theta}{k}$, we apply a standard probability amplification technique. That is, we repeat the sampling procedure independently for $R = O\left(\frac{9^d}{(1-2\alpha)^2} \ln \frac{k}{\theta}\right)$ iterations. By the union bound, this ensures that each event $\mathcal{E}_i$ holds for all $i \in [k]$ with total failure probability at most $\theta$. Each iteration of the algorithm involves computing line-to-point distances and sorting the lines based on proximity to candidate centers, which together require $O(n \log n)$ time. Therefore, the total running time of the high-dimensional algorithm becomes $O\left(\frac{9^d}{(1-2\alpha)^2} \cdot n \log n \cdot \ln \frac{k}{\theta}\right)$. Combining this run-time bound with the $(1 + O(\alpha))$-approximation guarantee established in the previous sections, we conclude the proof of Theorem 2.

## 4 EXPERIMENTAL RESULTS

In this section, we conduct empirical evaluations to assess the performance of the proposed algorithms. All methods were implemented in Python and executed on a machine with an Intel Core i7-14700KF processor and 32 GB RAM. Following the prior work (Nguyen et al. (2023); Ergun et al. (2022); Huang et al. (2025)), we run each algorithm 10 times and report the average results with deviations.

Table 2: Experimental Comparisons between Co-SA and LA-$k$ML.

| Dataset | Method | n/d/k | Min_cost | Max_cost | Mean_cost | Std | Time |
|---------|--------|-------|----------|----------|-----------|-----|------|
| Syn 1 | Co-RS | 1000/5/3 | 1.80E+03 | 2.39E+03 | 2.03E+03 | 1.81E+02 | 2.31E+02 |
|        | LA-$k$ML | | **1.69E+03** | **1.75E+03** | **1.72E+03** | **2.58E+01** | **8.16E+01** |
| Syn 2 | Co-RS | 5000/10/3 | | | | | over 24 hours |
|        | LA-$k$ML | | 8.43E+03 | 8.56E+03 | 8.49E+03 | 6.50E+01 | 2.36E+04 |
| Syn 3 | Co-RS | 10000/5/3 | | | | | over 24 hours |
|        | LA-$k$ML | | 1.71E+04 | 1.73E+04 | 1.72E+04 | 1.16E+02 | 8.54E+04 |
| Road | Co-RS | 418/2/3 | 2.79E+00 | 3.55E+00 | 3.08E+00 | 3.40E-01 | 2.82E+00 |
|        | LA-$k$ML | | **1.52E+00** | **2.33E+00** | **1.80E+00** | **3.71E-01** | **2.32E+00** |

**Datasets.** Following the work in Marom & Feldman (2019), we evaluate our algorithms on both synthetic and real-world datasets. The synthetic datasets are generated under carefully controlled parameters, which allow systematic variation of both the dimensionality (e.g., $d = 2, 5, 10$) and the input size, thereby facilitating a comprehensive evaluation of the scalability and robustness of the algorithms. The real dataset is obtained from the Open Street Map road network (denoted as Road). In this dataset, each road is modeled as a two-dimensional segment that is further extended to an infinite line in the plane.

**Algorithms.** In our experiments, we primarily compare the proposed LA-$k$ML algorithm with the coreset construction algorithm combined with random sampling (denoted as Co-RS) introduced by Marom & Feldman (2019). For the LA-$k$ML algorithm, we set the error rate $\alpha$ to 0.5, and fix $R = 5$. For the Co-RS algorithm, we set the sampling size to $0.2n$. We also conduct parameter sensitivity experiments to evaluate the robustness of our algorithm with respect to key hyperparameters (see Appendix A.6 for the proof).

**Predictor Description.** Following the prior work (Ergun et al. (2022)), the predictor is generated as follows. For each dataset, we first run the Co-RS algorithm (Marom & Feldman (2019)) method as an initialization, where the labels returned are regarded as the optimal labeling partitions. To test the performance of the algorithms under different error rates of the predictor, following the previous work of Ergun et al. (2022), we randomly change the labels of the $\alpha m_i$ points closest to $c_i$ for each cluster $P_i$ to generate the corrupted labeling partitions $L'_1, \ldots, L'_k$ as the predictors.

**Results.** Table 3 compares Co-RS with our proposed LA-$k$ML algorithm on both synthetic and real datasets under different input sizes and dimensions. "Co-RS" denotes the sampling-based baseline without predictive information, while "LA-$k$ML" represents our learning-augmented approach. Reported values include the minimum, maximum, mean, and standard deviation of clustering costs, along with the corresponding running times. Reported values include the minimum, maximum, mean, and standard deviation of clustering costs, along with the corresponding running times. For large-scale instances, LA-$k$ML fails to terminate within 24 hours, which highlights the computational advantage of our method in low-dimensional settings.

## 5 CONCLUSIONS

In this paper, we consider the learning-augmented algorithm for the $k$-median of lines problem. This problem generalizes classical point-based clustering by requiring center selection for sets of lines rather than points, introducing significant geometric and algorithmic challenges. To address these difficulties, we propose the Anchor Set construction, a new geometric structure that defines bounded regions via sampled line pairs, enabling robust candidate selection without relying on the triangle inequality. Based on this technique, we design a sampling algorithm that leverages predicted cluster labels with bounded error to guide the center selection process. Our algorithm achieves a $(1+O(\alpha))$-approximation guarantee, where $\alpha$ denotes the predictor's error rate. Notably, in low-dimensional settings, the algorithm runs in near-linear time with respect to the number of lines, making it practical for large-scale applications.

## ETHIC STATEMENT

This work makes purely methodological and theoretical contributions to clustering algorithms. It does not involve human subjects, personal or sensitive data, or any experiments with potential negative societal impact. The datasets used are publicly available benchmark datasets, and no ethical concerns are associated with their use.

## REPRODUCIBILITY STATEMENT

We have made extensive efforts to ensure the reproducibility of our results. All theoretical claims are stated with precise assumptions and supported by complete proofs in the appendix. The algorithms are described in detail in the main text, with pseudocode included for clarity. Experimental settings, parameter choices, and evaluation protocols are clearly discussed in both the main context and appendix. We use only publicly available benchmark datasets, and the data processing steps are fully described in the main context and appendix.

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

# A    MISSING PROOFS

## A.1    PROOF OF LEMMA 6

**Lemma 6.** *For any $i \in [k]$, we have $L_i \setminus L_i' = ((L_i \cap L_i^*) \setminus A_i) \cup B_i$ .*

*Proof.* The lemma follows directly from standard set operations. Indeed,

$$
L_i \setminus L_i' = ((L_i \setminus L_i') \cap L_i^*) \cup ((L_i \setminus L_i') \setminus L_i^*) = ((L_i \setminus L_i') \cap L_i^*) \cup (L_i \setminus (L_i' \cup L_i^*))
$$
$$
= ((L_i \setminus L_i') \cap L_i^*) \cup B_i.
$$
$$
\Rightarrow (L_i \setminus L_i') \cap L_i^* = (L_i \setminus L_i') \setminus B_i.
$$

Also note that $L_i \cap L_i^* = ((L_i \cap L_i^*) \cap L_i') \cup ((L_i \cap L_i^*) \setminus L_i')$. Therefore,

$$
(L_i \cap L_i^*) \setminus L_i' = (L_i \cap L_i^*) \setminus ((L_i \cap L_i^*) \cap L_i') = (L_i \cap L_i^*) \setminus A_i.
$$

Since $(L_i \setminus L_i') \cap L_i^* = (L_i \cap L_i^*) \setminus L_i'$, we have $(L_i \setminus L_i') \setminus B_i = (L_i \cap L_i^*) \setminus A_i$. Thus, we conclude that $L_i \setminus L_i' = ((L_i \cap L_i^*) \setminus A_i) \cup B_i$. $\qquad\square$

## A.2    PROOF OF LEMMA 8

**Lemma 8.** *Let $R = \frac{4}{t(1-2\alpha)^2} \ln \frac{k}{\theta}$. Then, after $R$ repetitions, with probability at least $1 - \frac{\theta}{k}$, we have $\Delta(L_i \setminus L_i', \hat{c}_i) \leq (1 + 8\alpha)\tau_i$ .*

*Proof.* Recall that the event $\mathcal{E}_i$ occurs with probability at least $\frac{t(1-2\alpha)^2}{4}$ in each iteration. Hence, the complementary event (i.e., $\mathcal{E}_i$ does not occur) happens with probability at most $1 - \frac{t(1-2\alpha)^2}{4}$. Consequently, the probability that $\mathcal{E}_i$ never occurs across all $R$ repetitions is at most $(1 - \frac{t(1-2\alpha)^2}{4})^R$. For $R = \frac{4}{t(1-2\alpha)^2} \ln \frac{k}{\theta}$, we bound this probability as follows $(1 - \frac{t(1-2\alpha)^2}{4})^R \leq e^{-\frac{t(1-2\alpha)^2}{4} \cdot R} \leq e^{-\ln \frac{k}{\theta}} \leq \frac{\theta}{k}$. Thus, with probability at least $1 - \frac{\theta}{k}$, the event $\mathcal{E}_i$ occurs at least once, implying $\Delta(L_i \setminus L_i', \hat{c}_i) \leq (1 + 8\alpha)\tau_i$. $\qquad\square$

## A.3    PROOF OF LEMMA 9

**Lemma 9.** *If $\Delta(L_i \setminus L_i', \hat{c}_i) \leq (1 + 8\alpha)\tau_i$, then it holds that $\|\hat{c}_i - c_i^*\| \leq \frac{(2+8\alpha)\tau_i}{(1-2\alpha)n_i}$.*

*Proof.* By the assumption, for any line $\ell \in L_i^* \cap (L_i \setminus L_i')$, we have $\|\hat{c}_i - c_i^*\| \leq \delta(\hat{c}_i, \ell) + \delta(c_i^*, \ell)$. Summing this inequality over all lines $\ell \in L_i^* \cap (L_i \setminus L_i')$, we get $|L_i^* \cap (L_i \setminus L_i')| \cdot \|\hat{c}_i - c_i^*\| \leq \sum_{\ell \in L_i^* \cap (L_i \setminus L_i')} \delta(\hat{c}_i, \ell) + \sum_{\ell \in L_i^* \cap (L_i \setminus L_i')} \delta(c_i^*, \ell) \leq (1 + 8\alpha)\tau_i + \tau_i \leq (2 + 8\alpha)\tau_i$. Since $|L_i^* \cap (L_i \setminus L_i')| \geq |L_i \cap L_i^*| - |A_i| \geq (1-\alpha)n_i - \alpha n_i = (1-2\alpha)n_i$, it follows that $\|\hat{c}_i - c_i^*\| \leq \frac{(2+8\alpha)\tau_i}{(1-2\alpha)n_i}$. $\qquad\square$

## A.4    PROOF OF LEMMA 10

**Lemma 10.** *With probability at least $1 - \frac{\theta}{k}$, the following inequality holds $\Delta(L_i \cap L_i^*, \hat{c}_i) \leq \Delta(L_i \cap L_i^*, c_i^*) + \frac{(8\alpha+32\alpha^2)\tau_i}{(1-2\alpha)}$ (see Appendix A.4 for the proof).*

*Proof.* Recall that with probability at least $1 - \frac{\theta}{k}$, we have $\|\hat{c}_i - c_i^*\| \leq \frac{(2+8\alpha)\tau_i}{(1-2\alpha)n_i}$. By combining this with the event $\Delta(L_i \setminus L_i', \hat{c}_i) \leq (1 + 8\alpha)\tau_i$, which holds with probability at least $1 - \frac{\theta}{k}$, we see by the union bound that both conditions hold simultaneously with probability at least $1 - \frac{\theta}{k}$. Conditioning on these events, and noting that $L_i \cap L_i^* = ((L_i \setminus L_i') \setminus B_i) \cup A_i$, we can rewrite

$$
\Delta(L_i \cap L_i^*, \hat{c}_i) - \Delta(L_i \cap L_i^*, c_i^*)
$$
$$
= (\Delta(L_i \setminus L_i', \hat{c}_i) - \Delta(L_i \setminus L_i', c_i^*)) + (\Delta(B_i, c_i^*) - \tag{3}
$$
$$
\Delta(B_i, \hat{c}_i)) + (\Delta(A_i, \hat{c}_i) - \Delta(A_i, c_i^*)).
$$

Table 3: Impact of Hyperparameters on LA-$k$ML Performance

| Hyperparameter | value | Min_cost | Max_cost | Mean_cost | Std | Time |
|---|---|---|---|---|---|---|
| | 3 | 1.52E+00 | 2.33E+00 | 1.80E+00 | 3.71E-01 | 2.32E+00 |
| $k$ (Syn 1) | 5 | 2.29E+00 | 7.11E+00 | 3.65E+00 | 1.29E+00 | 2.19E+00 |
| | 10 | 3.75E+00 | 1.47E+01 | 6.02E+00 | 3.08E+00 | 1.30E+00 |
| | 5 | 1.69E+03 | 1.75E+03 | 1.72E+03 | 2.58E+01 | 8.16E+01 |
| $R$ (Road) | 10 | 1.71E+03 | 1.75E+03 | 1.73E+03 | 2.16E+01 | 6.19E+01 |
| | 20 | 1.71E+03 | 1.85E+03 | 1.78E+03 | 7.04E+01 | 3.27E+02 |

We now bound each term separately. First, consider the term involving $B_i$.

$$\Delta(B, c_i^*) - \Delta(B, \hat{c}_i) = \sum_{\ell \in B}(\delta(\ell, c_i^*) - \delta(\ell, \hat{c}_i)) \leq \sum_{\ell \in B_i}(2\|\hat{c}_i - c_i^*\|)$$

$$\leq 2\alpha n_i \cdot \frac{(2+8\alpha)\tau_i}{(1-2\alpha)n_i} = \frac{(4\alpha+16\alpha^2)\tau_i}{(1-2\alpha)}.$$

By a symmetric argument, we similarly bound the term involving $A_i$ $\Delta(A, c_i^*) - \Delta(A, \hat{c}_i) \leq \frac{(4\alpha+16\alpha^2)\tau_i}{(1-2\alpha)}$. Finally, noting from optimality of $c_i^*$ that $\Delta(L_i \cap L_i^*, \hat{c}_i) - \Delta(L_i \cap L_i^*, c_i^*) \leq 0$. Equation (3) is bounded by $\frac{(4\alpha+16\alpha^2)\tau_i}{(1-2\alpha)} + \frac{(4\alpha+16\alpha^2)\tau_i}{(1-2\alpha)} = \frac{(8\alpha+32\alpha^2)\tau_i}{(1-2\alpha)}$. $\qquad\square$

## A.5 PROOF OF LEMMA 11

**Lemma 11.** *With probability at least* $1 - \frac{\theta}{k}$, $\Delta(L_i^*, \hat{c}_i) \leq (1 + \frac{12\alpha+40\alpha^2-32\alpha^3}{(1-\alpha)(1-2\alpha)})\tau_i$.

*Proof.* First, it is easy to get that $\Delta(L_i^*, \hat{c}_i) = \Delta(L_i^* \setminus L_i, \hat{c}_i) + \Delta(L_i^* \cap L_i, \hat{c}_i)$. We bound the first term using the triangle inequality. For each line $\ell \in L_i^* \setminus L_i$, we have $\delta(\ell, \hat{c}_i) \leq \delta(\ell, c_i^*) + 2\|\hat{c}_i - c_i^*\|$. Summing over all lines in $L_i^* \setminus L_i$, and noting $|L_i^* \setminus L_i| \leq \frac{\alpha n_i}{1-\alpha}$, we obtain

$$\Delta(L_i^* \setminus L_i, \hat{c}_i) = \sum_{\ell \in L_i^* \setminus L_i} \delta(\ell, \hat{c}_i) \leq \sum_{\ell \in L_i^* \setminus L_i}(\delta(\ell, c_i^*) + 2\|\hat{c}_i - c_i^*\|)$$

$$= \Delta(L_i^* \setminus L_i, c_i^*) + \frac{2\alpha n_i(2+8\alpha)\tau_i}{(1-\alpha)(1-2\alpha)n_i}$$

$$= \Delta(L_i^* \setminus L_i, c_i^*) + \frac{2\alpha(2+8\alpha)\tau_i}{(1-\alpha)(1-2\alpha)}.$$

Next, we bound the second term. By Lemma 10, with probability at least $1 - \frac{\theta}{k}$, we have

$$\Delta(L_i \cap L_i^*, \hat{c}_i) \leq \Delta(L_i \cap L_i^*, c_i^*) + \frac{(8\alpha+32\alpha^2)\tau_i}{(1-2\alpha)}.$$

Combining both bounds, we have

$$\Delta(L_i^*, \hat{c}_i) = \Delta(L_i \cap L_i^*, c_i^*) + \frac{(8\alpha+32\alpha^2)\tau_i}{(1-2\alpha)} + \Delta(L_i^* \setminus L_i, c_i^*) + \frac{2\alpha(2+8\alpha)\tau_i}{(1-\alpha)(1-2\alpha)}$$

$$\leq \Delta(L_i^*, c_i^*) + \frac{(8\alpha-8\alpha^2+32\alpha^2-32\alpha^3+4\alpha+16\alpha^2)\tau_i}{(1-\alpha)(1-2\alpha)}$$

$$= \Delta(L_i^*, c_i^*) + \frac{(12\alpha+40\alpha^2-32\alpha^3)\tau_i}{(1-\alpha)(1-2\alpha)} = (1 + \frac{12\alpha+40\alpha^2-32\alpha^3}{(1-\alpha)(1-2\alpha)})\tau_i.$$

$\qquad\square$

## A.6 HYPERPARAMETER ANALYSIS FOR LA-$k$ML

Table 3 reports the impact of hyperparameters on the performance of LA-$k$ML. For the number of clusters $k$, smaller values yield lower and more stable costs, while larger $k$ improves model expressiveness at the expense of higher costs and variance. For the parameter $R$, increasing its value has only a marginal effect on improving solution quality, but significantly increases the running time. Overall, these results suggest that careful tuning of $k$ and $R$ is essential to balance clustering accuracy and computational efficiency.

### A.7 LLM USAGE

Large language models (LLMs) were used only for minor language polishing and did not contribute to the research content.

