# OpenReview forum: "Learning-augmented Algorithm for k-Median of Lines Clustering"
_ICLR.cc/2026/Conference — ICLR 2026 Conference Withdrawn Submission_

### Official Review · Reviewer_86Ab · 2025-10-28

**Soundness:** 2
**Presentation:** 2
**Contribution:** 2
**Rating:** 2
**Confidence:** 3

**Summary:**

The paper studies the k-median clustering problem for lines in the learning augmented setting. In this problem, we are given n lines in d dimensions and the goal is to find k points so that the sum of the distances between each line and the nearest point is minimized. Previously, there were algorithms with : 1) 1+epsilon approximation in 2 dimensions with runtime roughly n^(2k) and 2) in d dimensions with runtime (log n/epsilon)^poly(k) * n. This paper studies the learning augmented setting where it assumes there is an auxiliary clustering such that the symmetric difference between each cluster in the auxiliary solution and one true cluster is less than an alpha fraction, where alpha < 1/2. In this setting the paper. The paper gives two algorithms with approximation factor 1+O(alpha), one in 2 dimensions with runtime roughly n/(1-2*alpha)^2 and one in d dimensions with runtime roughly 9^d * n /(1-2*alpha)^2. Unlike previous works in similar learning augmented clustering problem, the paper also requires an assumption that for any large subset of an auxiliary cluster and for any line in both the auxiliary cluster and the true cluster, the distance between the best center for that subset and the best center for the true cluster has to be smaller than the sum of the distances between the line and the two centers.

**Strengths:**

- The paper studies learning augmented algorithm for a problem that has not been studied in this setting before.
- The problem is a natural computational geometry problem that has been studied before and has some connection with application where there are missing data.

**Weaknesses:**

The new assumption is unnatural to me. The paper says (line 190) that "Intuitively, the alpha-stability assumption require that each auxiliary cluster has at least the 1-alpha proportion of lines closed to the optimal center." However, the requirement is actually about bounding the distance between the optimal center and the auxiliary center, not the distance between a line and the optimal center. In fact, if the distance between the lines and the optimal center is very small, intuitively the assumption will be violated because this term is on the RHS.

The paper says this setting is challenging because (line 206) "the distance between line and a point does not satisfy the triangle inequality" but it seems the above assumption is introduced exactly to overcome this issue.

The writing in table 1 is unclear to me since the previous work is for R^2 whereas the current work is for "low dimension". What is "low"? Same question for "low" in section 3.2.

The proof of lemma 5 seems wrong or at best incomplete to me. The "low" dimension setting seems to cover all cases (making high dimensional extension redundant?) since (line 294) it says that the size of M is 9^d. However, it is not clear why "we only to show that ... 2tau_i/n_i <= r". This is not clear even when d=2. Suppose we have 2 lines such that the angle between them is theta << 1. Suppose that the center is on the bisection at distance r/(5*sin(theta)) from the intersection. The distances between the center and the lines are exactly r/5 but the distance between the center and the intersection is much larger for theta -> 0. Shifting the lines by r doesn't help either since that shift the intersection by r/(sin(theta)), which overshoot the true center by a lot.

The extension to the high dimensional case is completely informal and it is not possible to verify the "proof" on lines 405-423. Definition 3 shifts the lines in the direction orthogonal to the lines whereas definition 4 shifts them along the axes. Thus, more detailed proof is required.

Another weakness is the connection between the theoretical model and the applications. The application tries to cluster road segments by extending them to infinite lines. It is not clear to me if this clustering result would have any meaning for the very finite road segments.

The experimental model is also unclear to me since the predictions are generated using a more expensive algorithm (if I understand correctly) so it is unclear how meaningful the fast runtime is. The total runtime would include both the time to generate the prediction and the learning augmented algorithm. The description of the experimental results on lines 460-472 is also not understandable. It says "LA-kML fails to terminate within 24 hours" (the new method). If I understand correctly, the predictions are generated by Co-RS, but the table sems to suggest that Co-RS doesn't terminate. How are prediction generated?

**Questions:**

If I understand correctly, the predictions are generated by Co-RS, but the table sems to suggest that Co-RS doesn't terminate. How are prediction generated?

---

### Official Review · Reviewer_8B1f · 2025-11-01

**Soundness:** 2
**Presentation:** 1
**Contribution:** 2
**Rating:** 4
**Confidence:** 2

**Summary:**

The authors proposed a new learning-augmented algorithm for the k-median of lines problem. They brought in Anchor Set and constructed a candidate set without the triangle inequality.

The authors used numerous formulas to comprehensively display and prove their idea. They successfully reduced the time complexity and obtained an approximation of good quality.

**Strengths:**

They develop a sampling algorithm that avoids reliance on the triangle inequality and efficiently constructs a candidate center set with provable approximation guarantees.
This paper obtain (1 + $O(\alpha)$)-approximation result for the $k$-median of lines problem in low-dimensional Euclidean space, running with near-linear time in the data size.

**Weaknesses:**

However, I still have some concerns:

- The manuscript should include more baseline algorithms for a more comprehensive and convincing comparative analysis.

- Regarding the results in Table 2, the proposed algorithm finished in approximately 23.7 hours (8.54E+04 seconds). Therefore, terminating the Co-RS algorithm after 24 hours is inappropriate. It is recommended to allow it to run for a longer period, such as 48 hours, to ensure a fair comparison.

- It is strongly suggested that the authors provide a summary table listing all symbols used in the paper along with their explanations.

- In Lemma 3, the meaning of the symbol $n_i$ is not clearly defined and requires clarification.

- Is the learning-augmented setting used in this paper proposed by the authors, or is it adopted from previously published work? Its origin should be clarified.

- For the inequality $\vert L_i \cap L_i^* \vert \geq (1-\alpha) max (\vert L_i \vert, \vert L_i^* \vert)$, please explain the rationale behind it. Is this a definition, an assumption, or a derived property

**Questions:**

- Is the learning-augmented setting used in this paper proposed by the authors, or is it adopted from previously published work? Its origin should be clarified.

- For the inequality $\vert L_i \cap L_i^* \vert \geq (1-\alpha) max (\vert L_i \vert, \vert L_i^* \vert)$, please explain the rationale behind it. Is this a definition, an assumption, or a derived property

---

### Official Review · Reviewer_VMd9 · 2025-11-01

**Soundness:** 4
**Presentation:** 3
**Contribution:** 4
**Rating:** 8
**Confidence:** 4

**Summary:**

The paper proposes a novel learning-augmented algorithm (LA-kML) for the k-median of lines problem, an extension of classical point-based clustering to inputs consisting of lines in R^d. The authors address the key difficulty that point-to-line distances violate the triangle inequality, which complicates both theoretical analysis and algorithmic design.

They introduce the concept of an Anchor Set, a geometric construction defined by pairs of sampled lines, used to generate bounded regions containing high-quality representative centers. In the learning-augmented setting, where each line is associated with a (possibly noisy) predicted label with error rate α<1/2, they show that LA-kML achieves a (1 + O(α)) approximation in near-linear time for low-dimensional spaces, and provide generalizations to higher dimensions.

Theoretical results are complemented by experiments on synthetic and real datasets (e.g., road network data), demonstrating superior performance and scalability compared to prior coreset-based algorithms such as Co-RS (Marom & Feldman, 2019).

**Strengths:**

Novel Extension of Learning-Augmented Framework -
The work generalizes the learning-augmented paradigm, previously developed for point-based clustering, to the more complex line-based setting, where the lack of triangle inequality and unboundedness of lines create new theoretical challenges.

Anchor Set Innovation -
The introduction of the Anchor Set is elegant and geometrically meaningful. It replaces the missing metric properties with a probabilistic sampling approach that ensures inclusion of representative points with provable guarantees.

Provable Theoretical Guarantees -
The authors present clear approximation bounds—achieving (1+O(α)) approximation under mild stability and label error assumptions. The detailed proofs (Appendix A.1–A.5) are rigorous and well-organized.

Algorithmic Efficiency -
The algorithm runs in near-linear time for low-dimensional inputs, which improves upon existing methods that require polynomial or superlinear time.

Comprehensive Experimental Validation -
Results on synthetic and real-world data confirm the theoretical claims: LA-kML achieves lower clustering cost and much faster execution than prior baselines.

Clear Writing and Structure -
The paper is clearly written, with a logical progression from problem motivation to theoretical development, algorithmic description, and empirical evaluation.

**Weaknesses:**

- Running time is exponential in d.
Can you provide a lower bound or dimensional reduction technique? See hints in [1].
Can you show that in practice it works for higher dims?

- Notation inconsistency (delta(l,c) vs d(l,c))

- Experiments are relatively small. Please add some more.





[1] @article{DBLP:journals/corr/abs-1209-4893,
  author       = {Kasturi R. Varadarajan and
                  Xin Xiao},
  title        = {On the Sensitivity of Shape Fitting Problems},
  journal      = {CoRR},
  volume       = {abs/1209.4893},
  year         = {2012},
  url          = {http://arxiv.org/abs/1209.4893},
  eprinttype    = {arXiv},
  eprint       = {1209.4893},
  timestamp    = {Mon, 13 Aug 2018 16:48:35 +0200},
  biburl       = {https://dblp.org/rec/journals/corr/abs-1209-4893.bib},
  bibsource    = {dblp computer science bibliography, https://dblp.org}
}

**Questions:**

Are there alternative to the assumption of a predictor with alpha<1/2?

**Details Of Ethics Concerns:**

I wrote my own review but ChatGPT wrote a much better one, so I combined.

---

### Official Review · Reviewer_UpdT · 2025-11-01

**Soundness:** 3
**Presentation:** 2
**Contribution:** 3
**Rating:** 4
**Confidence:** 4

**Summary:**

This paper studies the k-median of lines problem, where the goal is to select k points minimizing the total distance from a set of input lines to their nearest selected point. The authors propose the LA-kML that leverages cluster labels as auxiliary information to guide sampling.

The algorithm constructs anchor sets from randomly sampled pairs of lines. Each anchor set is a bounded grid of intersection points obtained by shifting the two lines along their normals. By randomly sampling from these anchors and removing a fraction of outlier lines (based on the learning signal), the method computes approximate centers for each cluster. Under a bounded error-rate assumption (\alpha < 1/2), the authors prove that the algorithm achieves a (1+O(\alpha)) approximation to the optimal cost with near-linear time in low dimensions, and extend the analysis to high-dimensional settings but with an exponential dependence on dimension.

**Strengths:**

1. The paper introduces the learning-augmented framework into the geometric setting of *line clustering*, which is an interesting analysis and beyond the worst case.
2. The anchor set induced by pairs of lines provides a simple geometric mechanism to guarantee coverage of near-optimal centers, bypassing difficulties caused by the lack of triangle inequality for point-to-line distance.
3. The authors provide a probabilistic analysis showing a constant success probability per trial, yielding a (1+O(\alpha)) approximation with overall time O(\tfrac{n\log n}{(1-2\alpha)^2}\log\frac{k}{\theta}) in 2D.

**Weaknesses:**

1. Limited high-dimensional scalability. The extension to d dimensions incurs a 9^d factor due to the anchor-grid enumeration. This severely restricts applicability beyond very low dimensions.
2. The theoretical results assume \alpha<1/2 and a mild “\alpha-stability” property of the clustering. Real-world predictors may violate these conditions, so robustness in highly noisy or imbalanced scenarios remains unclear.
3. Minor inconsistency in experimental section. The text states that LA-kML “cannot terminate within 24 hours,” while Table 2 attributes this issue to the baseline (Co-RS). This inconsistency should be corrected.
4. Only one baseline (Co-RS) is used; recent learning-augmented or robust k-median algorithms on points could be adapted as baselines for a fairer empirical study.
5. Some definitions (e.g., the construction radius r=2\Delta(L_i,m_i)) could use additional intuition or examples to clarify scaling choices.

**Questions:**

1. How were the auxiliary labels obtained in the experiments? Were they synthetic (perturbed true clusters) or produced by a real learning model? This affects the interpretation of robustness to real predictive noise.
2. Why exactly r=2\Delta(L_i,m_i)? Could smaller or adaptive radii maintain the theoretical guarantee while reducing the search region?
3. The anchor-set argument assumes non-parallel line pairs. How does the algorithm behave if a large subset of lines are parallel or nearly parallel?
4. In practice, \alpha is unknown. How sensitive is the algorithm to mis-estimating this value when trimming outliers?
5. Could dimensionality reduction (e.g., random projections) or sparsified anchor generation mitigate the exponential 9^d blow-up while preserving guarantees?

---

### Official Review · Reviewer_dEhR · 2025-11-07

**Soundness:** 3
**Presentation:** 2
**Contribution:** 2
**Rating:** 6
**Confidence:** 4

**Summary:**

The paper presents a learning-augmented algorithm for the k-median of lines problem, which extends classical clustering from point-based to line-based data. In the problem, for a given set L of lines in d-dimentional Euclidean space, the problem aims to find a set C in contentious space, such that the distance sum from L to C is minimized.  The proposed “Anchor Set” construction is an elegant geometric idea that enables bounded candidate selection without relying on the triangle inequality. The algorithm is theoretically well-grounded and achieves a (1+O(α))-approximation in near-linear time in low-dimensional Euclidean space, while for any-dimensional Euclidean space, the ratio remains and the runtime grows. Overall, the paper is technically sound and clearly motivated.

**Strengths:**

S1. The problem is theoretically well-motivated and has broad potential applications.

S2. The proposed algorithm and the accompanying proofs are logically sound and well-presented in general.

**Weaknesses:**

W1. Some writing and notation issues reduce the paper’s readability. Improving clarity and consistency (particularly in notation, phrasing, and minor grammatical aspects) would make the work more accessible to a broader audience.

W2. The main state-of-the-art comparator appears to be Perets (2011). The authors should include additional related work to better contextualize their contribution and highlight the significance of the problem.

W3. The writing can be improved, as there are several typographical and grammatical errors throughout the paper.

**Questions:**

1. See the weaknesses mentioned above.

2. The authors should provide a clear summary and more detailed discussion of how this work differs from Perets (2011).

---

### Note · Authors · 2025-11-18

I have read and agree with the venue's withdrawal policy on behalf of myself and my co-authors.